# Loco-Regional Therapies in Oligometastatic Adrenocortical Carcinoma

**DOI:** 10.3390/cancers14112730

**Published:** 2022-05-31

**Authors:** Charles Roux, Alice Boileve, Matthieu Faron, Livia Lamartina, Alexandre Delpla, Lambros Tselikas, Jérome Durand-Labrunie, Segolène Hescot, Thierry de Baere, Julien Hadoux, Frederic Deschamps, Eric Baudin

**Affiliations:** 1Gustave Roussy, Département de Radiologie Interventionnelle, F-94805 Villejuif, France; adelpla@ghpsj.fr (A.D.); lambros.tselikas@gustaveroussy.fr (L.T.); thierry.debaere@gustaveroussy.fr (T.d.B.); frederic.deschamps@gustaveroussy.fr (F.D.); 2Gustave Roussy, Département D’oncologie Endocrinienne, F-94805 Villejuif, France; alice.boileve@gustaveroussy.fr (A.B.); livia.lamartina@gustaveroussy.fr (L.L.); julien.hadoux@gustaveroussy.fr (J.H.); eric.baudin@gustaveroussy.fr (E.B.); 3Gustave Roussy, Département de Chirurgie, F-94805 Villejuif, France; matthieu.faron@gustaveroussy.fr; 4Gustave Roussy, Département de Radiothérapie, F-94805 Villejuif, France; jerome.durand-labrunie@gustaveroussy.fr; 5Institut Curie, Département de Médecine Oncologique, F-75005 Paris, France; segolene.hescot@curie.fr

**Keywords:** adrenocortical carcinoma, loco-regional treatments, oligometastatic, prognosis factors, interventional radiology

## Abstract

**Simple Summary:**

The treatment recommended for stage IVa Adrenocortical carcinoma (ACC) not amenable to radical resection is the mitotane plus loco-regional treatment (LR) strategy, which has not yet been validated. Moreover, prognosis factors for this strategy are not yet established. This study aimed to determine which stage IVa ACC patient population would benefit the most from this association. For this purpose, we reviewed all stage IVa patients (≤2 tumoral organs) treated with mitotane and LR from 2008 to 2021 in our institution. This study included 60 patients and 109 LR were performed. The primary endpoint was disease control (DC). We found that DC was associated with longer Time to second line Treatments (TTC). Moreover, DC rate was higher in patients that had ≤5 metastases or a maximum metastasis diameter below 3 cm. Based on those results we propose the first definition of oligometastatic ACC: stage IVa patients with ≤5 metastases or a maximum metastasis diameter below 3 cm. It is vitally important that scientists are able to describe their work simply and concisely to the public, especially in an open-access on-line journal.

**Abstract:**

Objective: The recommended first-line treatment for low-tumor-burden ACC (stage IVa ACC) not amenable to radical resection is mitotane in association with loco-regional treatments (LRs). The aim of this study was to determine the patient population that would benefit the most from LR. Materials and methods: This retrospective monocentric expert center chart review study was performed from 2008 to 2021 and included stage IVa patients (≤2 tumoral organs) treated with LR (either radiotherapy, surgery, or interventional radiology). The primary endpoint was disease control (DC). Correlations between DC, time to systemic chemotherapy (TTC), overall survival (OS), and tumor characteristics were analyzed using Kaplan–Meier survival analysis and Cox’s proportional hazards regression model for multivariate analysis. Results: Thirty-four women (57%) and 26 men with a median age of 48.1 years (IQR: 38.3–59.8) were included. One hundred and nine LRs were performed, with a median of 2 (IQR: 1–3) per patient. DC was achieved in 40 out of 60 patients (66.7%). Patients with DC had a significantly longer TTC (HR: 0.27, *p* < 0.001) and OS (HR: 0.22, *p* < 0.001). Patients with less than or equal to 5 metastases (HR: 6.15 (95% CI: 1.88–20.0), *p* = 0.002) or a maximum metastasis diameter below 3 cm had higher rates of DC (HR: 3.78 (95% CI: 1.09–13.14), *p* = 0.035). Conclusion: stage IVa ACC patients with ≤5 metastases or a maximum metastasis diameter below 3 cm had favorable responses to LR. We propose the name oligometastatic ACC for this subgroup of patients.

## 1. Introduction

Adrenocortical carcinoma (ACC) is a rare and aggressive malignant tumor with an incidence of 0.7–2.0 new cases per million population per year and less than a 15% 5-year overall survival in the metastatic stage [1,2]. The prognosis of metastatic ACC is mainly driven by tumor burden and Grade–R status–Age–Symptom (GRAS) parameters [3]. Patients with distant metastatic disease (stage IV) represent 21–35% of ACC cases at diagnosis.

The new European Network for the Study of Adrenal Tumor Stage classification (mENSAT) allows a better prognostic stratification of patients with metastatic ACC [4,5], and since 2015, the combination of a modified ENSAT stage IV classification and GRAS parameters best specified the prognostic heterogeneity. Indeed, stage IVa patients (defined as those with metastatic disease with no more than 2 tumor sites) have a median OS of 21.2 months and a 5-year overall survival range of 0 to 55% depending on favorable or unfavorable GRAS parameters [3]. In a retrospective study by Libé et al., 31% of metastatic ACC patients were stage IVa.

The treatment recommended for stage IVa ACC [6] not amenable to radical resection is the mitotane plus loco-regional treatment (LR) strategy, which has not yet been validated. Moreover, the second-line treatment is platin-based chemotherapy, which exhibits limited efficacy, and no validated third-line treatment is available. Most of the time, interventional radiology plays the lead role among LR because of its minimal invasiveness and the ability to administer repeated treatments over time. Preliminary studies have suggested that such a strategy is valid in selected patients and can be associated with long-term survival [7,8,9,10,11,12,13]. Boileve et al. [14] recently presented new results that highlighted an increase in time to chemotherapy initiation and overall survival in selected stage IVa ACC patients eligible for the LR strategy compared with patients treated with mitotane only at baseline.

The objective of this retrospective monocentric study was to identify prognostic factors of LR in stage IVa ACC patients in order to better specify the population that would benefit the most from this strategy.

## 2. Material and Methods

### 2.1. Patients and ACC Characteristics

This retrospective single-center study from a tertiary referral cancer center was approved by the Gustave Roussy Institutional Ethics Committee (n°: GR 2021-14). Written informed consent was obtained from each patient in accordance with the policy of our institution regarding chart reviews and the Declaration of Helsinki.

Between July 2003 and May 2018, all consecutive patients treated with mitotane for mENSAT stage IVa ACC (defined by the presence of a maximum of 2 tumor sites, including the primary tumor) were identified and followed until January 2021. The inclusion criteria were a confirmed diagnosis of ACC by expert pathologists, stage IVa ACC [15], tumor board decision to initiate LR, and available imaging during the follow-up period. The exclusion criteria were age under 18 years and the use of previous cytotoxic systemic treatments.

The following relevant data were collected from patient charts by two on-site investigators, AB and CR: age, sex, Weiss score, Ki67 percentage, mitotic count, primary surgery status (complete resection, R0, microscopic residual disease, R1, macroscopic residual disease, R2, resection not known, or Rx), and GRAS prognostic factors (age > 50 years, a Weiss score > 6 or a mitotic count > 20 and/or Ki67 > 20%, R1 or R2 status of adrenal surgery, and the presence of tumor-related or hormone-related symptoms). The disease-free interval (DFI) was defined as the duration between ACC diagnosis and stage IVa diagnosis, with DFI < 1 year representing synchronous metastasis and DFI ≥ 1 year representing metachronous metastasis. On the basis of previous studies (16–19) [16,17], the metastatic burden was categorized into subgroups according to the number of metastases (patients with more than 5 metastases were compared with patients with 5 metastases or less), metastasis maximal diameter (Dmax ≥ 3 cm or < 3 cm), and the number of metastatic organs.

### 2.2. Treatments and Complications

LR was decided by the tumor board by consensus among neuroendocrine oncologists, surgeons, radiotherapists, and interventional radiologists. LR characteristics and locations were collected; procedures included either surgery, stereotactic body radiation therapy (RT), chemoembolization (CHE), or percutaneous ablation (PA). The median number of procedures per patient was reported. Because of the retrospective nature of the study, only grade 3–5 adverse events according to Common Terminology Criteria for Adverse Events (CTCAE V.4) [18] were reported and categorized.

### 2.3. Best Response to Treatments and Follow-Up

The best response to the LR strategy during the follow-up was defined as follows: regardless of the number of LRs needed to reach it, complete response (CR) indicated the disappearance of all lesions; a partial response (PR) indicated at least a 30% decrease in the largest diameters of the lesions; and progressive disease (PD) indicated at least a 20% increase; otherwise, the tumor was classified as stable disease (SD). The patients whose best responses to LR during follow-up were complete response, partial response, or stable disease were defined as patients with controlled disease (DC). The time to progression after best response to LR was collected. The time to best response from LR initiation was collected, as well as the median number of LRs needed to reach the best response during patient follow-up.

Patients were followed by thoracic and abdomen–pelvic computer tomography (CT) scans plus or minus fluorodeoxyglucose (FDG) positron emission tomography (PET) scans every 3 months. They were reviewed by a single investigator (CR) to assess the best response to LR during follow-up according to the RECIST 1.1 criteria [19].

### 2.4. TTC and OS

Mitotane initiation was defined as the time of initiation of mitotane in synchronous metastatic patients or as the time of stage IVa diagnosis in cases of metachronous metastatic patients already receiving mitotane as adjuvant treatment. Time to second-line chemotherapy initiation (TTC) was defined as the time from mitotane initiation until systemic polychemotherapy initiation or clinical trial inclusion. Overall survival (OS) was defined as the time from mitotane initiation to death or censored at the time of last news.

### 2.5. Statistical Analysis

The primary endpoint of this study was DC, as defined by a best response obtained at any point during follow-up of either CR, PR, or SD. Secondary endpoints were correlations between DC and median TTC and median overall survival as well as DC and tumor characteristics. Categorical variables were analyzed using a Fischer exact test, and continuous variables were analyzed using the Mann–Whitney U test.

TTC and OS were calculated by Kaplan–Meier analysis. The correlations between tumor characteristics and DC and between DC and TTC or OS were analyzed. If significantly associated with DC in the univariate analysis (log-rank test), they were further tested in multivariate analyses. Hazard ratios (HRs) and 95% confidence intervals (95% CIs) were estimated using Cox’s proportional hazards regression model, with the lowest risk group as the reference group. All tests were two-sided. The statistical analysis was conducted using JMP software version 16. Results were statistically significant if *p* < 0.05.

## 3. Results

### 3.1. Patients Characteristics

Between July 2003 and May 2018, 79 patients with stage IVa ACC were diagnosed and discussed by our adrenal tumor board. Thirty-four female and twenty-six male (60/358, 16%) ACC patients or 75.9% (60/79) of stage IVa ACC patients were considered eligible for LR, with a median age of 48.1 years old (IQR: 38.3–59.8) (Figure 1). The characteristics of patients and tumors at stage IVa diagnosis are detailed in Table 1. All patients underwent primary surgery. Local relapse was observed in 21 patients (21/60, 36.2%) patients. Fifty-five (55/60, 91.2%) had at least one GRAS pejorative prognostic factor. DFI was higher than 1 year in 37 patients (37/60, 61.6%). Thirty-six patients had five or fewer metastases (36/60, 60%), and 46 (46/60, 76.7%) had maximal diameters of <3 cm. Nineteen patients (19/60, 31.6%) had one tumor site and forty-one had (41/60, 68.3%) two.

### 3.2. Treatments and Complications

All patients received mitotane (60/60, 100%) and 41 (35/60, 58.3%) achieved mitotane >14 mg/L at the time of loco-regional treatment initiation.

One-hundred and nine LRs were performed with a median of two (range: 1–12) per patient. Thirty-four (34/60, 57%) patients had surgeries, fourteen (14/60, 23%) had adrenal lodge RT, three (3/60, 5%) had organ radiotherapy, twenty (20/60, 33%) had CHE, and thirty (50%) had PA. The locations of the metastasis according to loco-regional treatments are detailed in Table 2. Thirty-one patients (31/60, 51.7%) had interventional radiological treatments only, 1 (1/60, 1.6%) patient had RT only, 9 patients (9/60, 15%) had surgery only; 21 patients (21/60, 35%) experienced a combination of different type of loco-regional treatments.

The complication rate per procedure was 6.4% (7/109). Two post-procedure bleedings with spontaneous hemostasis were reported (CTCAE grade 3). One gastrointestinal fistula and one mild pancreatitis occurred after LR (CTCAE grade 3). CTCAE grade 4 complications were two post-procedural adrenal insufficiencies and one pleural hemorrhage requiring embolization and two units of packed red blood cells.

### 3.3. Best Response to Treatments and Follow-Up

Overall DC rate and progressive Disease (PD) rate was 66.7% (40/60). The best response to LR during follow-up was stable disease in 10% (6/60), partial response in 8.3% (5/60), and CR in 29 patients (29/60; 48.3%).

DC (either CR, PR, or SD) was achieved after a median of 1 LR (min: 1–max: 10) and at a median time of 16 months (IQR: 19–3) from mitotane and LR initiation.

The median durations from SD, PR, and CR to systemic therapy were 44 (95% CI: 8–not reached), 70 (95% CI: 13–not reached), and 97 (95% CI: 16–not reached) months, respectively; at the time of progression after LR, 18 were still stage IVa (18/60, 30%).

### 3.4. TTC and OS

The median OS and TTC were 68 (95% CI: 43–117) and 42 (95% CI: 15–100) months, respectively (Appendix A).

Patients with DC had significantly longer TTC (HR: 0.27 (95% CI: 0.13–0.54), log-rank *p* < 0.001) and OS (HR: 0.22 (95% CI: 0.10–0.47), log-rank *p* < 0.001) (Figure 2). The one-year, two-year and five-year survival rates were 93.4%, 87.3%, and 71.3%, respectively, in DC patients and 97.4%, 63.0%, and 31.5% in PD patients.

The characteristics of patients with DC are detailed in Table 3. The following factors were correlated with DC: patients with ≤5 metastases (HR: 6.15 (95% CI: 1.88–20.0), *p* = 0.002), Dmax < 3 cm (HR: 3.78 (95% CI: 1.09–13.14), *p* = 0.035). Both factors were independently associated with DC (*p* = 0.006; *p* = 0.050, respectively, Table 4).

Finally, patients with DC had less cytotoxic chemotherapy within 6 months of mitotane initiation than patients with PD (*p* < 0.001) and during overall follow-up (*p* = 0.025).

## 4. Discussion

In the absence of progress in the field of systemic options, a pragmatic renewed interest has been observed for mitotane therapy optimization and locoregional therapy implementation in metastatic ACC patients [20,21,22]. The grounds for such interest in LR are three-fold: the high rate of response that can be achieved per lesion with favorable consequences on secretory control, the delayed antitumor activity of mitotane therapy, and the progress in the field of metastatic ACC prognostic stratification. Indeed, the identification of subgroups of ACC patients with more indolent courses allows a timely loco-regional sequential strategy to be implemented. Obviously, this strategy focuses on selected patients, 16% of ACC patients or 79% of stage IVa ACC patients in our center, with better prognosis prior to treatment, as first identified in a recent ENSAT publication [3]. Our study allows progress in the definition of the best candidates for LR and in the behavior of this subgroup of patients. On the basis of these results, we herein propose the first definition of oligometastatic ACC.

We hereby report a high DC rate of 66.6% (40/60), with the majority of best responses to LR strategy being complete response (48.3%). Importantly, LR in our study signifies the potential use, repetition, and combination of all available LR techniques as best determined during the tumor board meeting. This strategy is based on different profiles of indications and contraindications of each LR tool. In brief, surgery can access large lesions; IR is minimally invasive and can be repeated over time; and RT is able to treat infiltrative lesions. The varying number of LRs and time needed to reach DC can be explained by both the delay to achieve a therapeutic plasma mitotane level as well as the time to finalize the LR procedures and obtain the best possible response (Figure 3).

The median durations from SD, DC, and CR to untreatable progression with M + LR were 44, 70, and 97 months, respectively, emphasizing that a prolonged response can be achieved and systemic chemotherapy delayed. Interestingly, the best duration of DC was achieved with PR or even CR in 56.6% of patients. Such rates of PR or CR cannot be achieved by any systemic option, including mitotane alone, currently.

We looked for prognostic factors for DC with LR treatments in this homogenous population of metastatic ACC patients and identified two simple radiological parameters: five metastases at most or a maximum diameter below 3 cm are associated with higher rates of DC (HR: 6.15 and HR: 3.78, respectively). These results confirm the results of previous reports obtained in ACC populations and in other cancers, showing the importance of the maximum diameter of each metastasis and the number of metastases as a prognostic factor of response to several interventional radiology techniques. Surgery may be the ideal tool in cases of large but limited-number metastases. In case of an infiltrative lesion or a location at risk, radiotherapy could be the technique of choice. Overall, a combination of such techniques can be proposed and repeated over a patient’s treatment course (Figure 4). These results stress the importance of early minimally invasive interventions to expect the best response with a low complication rate, as previously reported by Cazejuste et al. [17] in 2010 and Velti et al. [23]. Interestingly, DC, including stabilization, was found to be independently associated with longer TTC and OS, suggesting that this endpoint could be a surrogate marker in future studies [24,25,26]. Overall, all stage IVa ACC patients for which either SD, PR, or CR is achievable through the combination and repetition of LR should be considered for the LR strategy.

There is a common effort among all oncologic specialties to provide specific definitions of oligometastatic disease for each cancer type. Hellman et al. [27] first defined oligometastatic cancer as “an intermediate stage between localized and systemic disease, where radical treatment of the primary cancer and of metastatic lesions might yield improved systemic control”. Since then, a more dynamic approach has been proposed on the basis of metachronous or synchronous evolution and response to treatment. Oligoprogression, oligorecurrence, and oligopersistence notions have emerged. By contrast, whether a precise number of metastatic lesions is relevant is debated and might differ between types of cancer depending on their specific biology [28,29,30,31]. In addition, a higher response rate to systemic options has been described after the reduction of the tumor burden [32,33]. Taken together, these proposals show that not only the disease course but also the response to treatments are taken into account to define the oligometastatic population. The multicentric “Stereotactic ablative radiotherapy versus standard of care palliative treatment in patients with oligometastatic cancers: (SABR-COMET)” phase II randomized trial [29] now serves as a proof-of-concept for the benefit of metastasis-guided LR in oligometastatic patients. In this trial, patients presented oligometastasis from a variety of primaries, including the colon, lung, breast, and prostate, and the sites of irradiated lesions included the lung, bone, liver, adrenal, and others. Patients in the SBRT group received stereotactic radiation to all sites of metastatic disease with the goal of achieving DC while minimizing potential toxicities. This trial reported a survival benefit of metastases-directed SBRT for oligometastatic patients (1–5 metastatic sites) who had their primary malignancy under control (median OS of 28 months in the control group, 95% CI: 19–33 vs. 41 months in the SBRT group, 95% CI: 26–not reached). Because of these interesting results, many trials are being conducted, among which SARON is one of the most ambitious [34]. Despite the growing number of studies and level of proof, few scientific societies propose recommendations for oligometastatic disease treatments: for NSCLC, the European Society of Medical Oncology (ESMO) still recommends protocol inclusion [35], and the National Cancer Comprehensive Network (NCCN) proposes radical treatments after careful tumor board assessment and in limited situations [36].

Libé et al. [3] demonstrated a significant difference in OS between stage IVa, IVb, and IVc patients on the basis of the tumor burden, a first step towards the definition of an oligometastatic ACC subgroup, but a more precise definition was still needed. On the basis of these results, we propose the name oligometastatic ACC for presentations of stage IVa ACC according to the modified ENSAT classification with five metastases at most or a maximum metastasis diameter below 3 cm. No genetic alteration has yet been identified as prognostic or predictive in metastatic ACC, but such investigation brings great expectations to identify oligometastatic ACC patients with prolonged courses early.

On the basis of our results, one may distinguish several possibilities in ACC patients: first, the targeting of all tumor sites at first treatment initiation, if achievable, and repeating LR until the best response is achieved; second, as has been previously suggested in other settings [32,33], in cases of oligoprogression, LR under mitotane therapy debulking aims to destroy mitotane-resistant metastasis to achieve global control. Overall, in both these possibilities, LR, among which IR is at the forefront, finds its place by achieving a high response rate and potentially repeatedly while allowing systemic treatment savings.

There are some limitations to this study, which include its retrospective and monocentric nature, the focus on a very selected subgroup of patients, the use of LR based on institution experience (high rate of interventional radiology treatments), and the low number of patients with bone metastases.

## 5. Conclusions

We identified a subgroup of stage IVa ACC patients according to modified ENSAT classification with specific behavior and favorable response to mitotane plus LR: patients with five metastasis at most or maximum diameter below 3 cm. On the basis of these simple radiological prognosis factors, we propose the name oligometastatic ACC for this subgroup of patients.

## Figures and Tables

**Figure 1 cancers-14-02730-f001:**
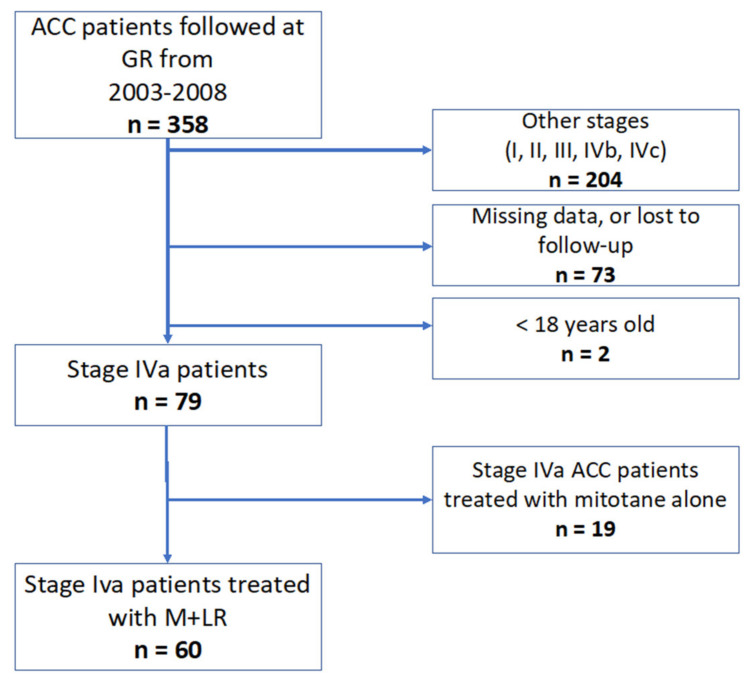
Flow-chart. ACC: adrenocortical carcinoma; GR: Gustave Roussy.

**Figure 2 cancers-14-02730-f002:**
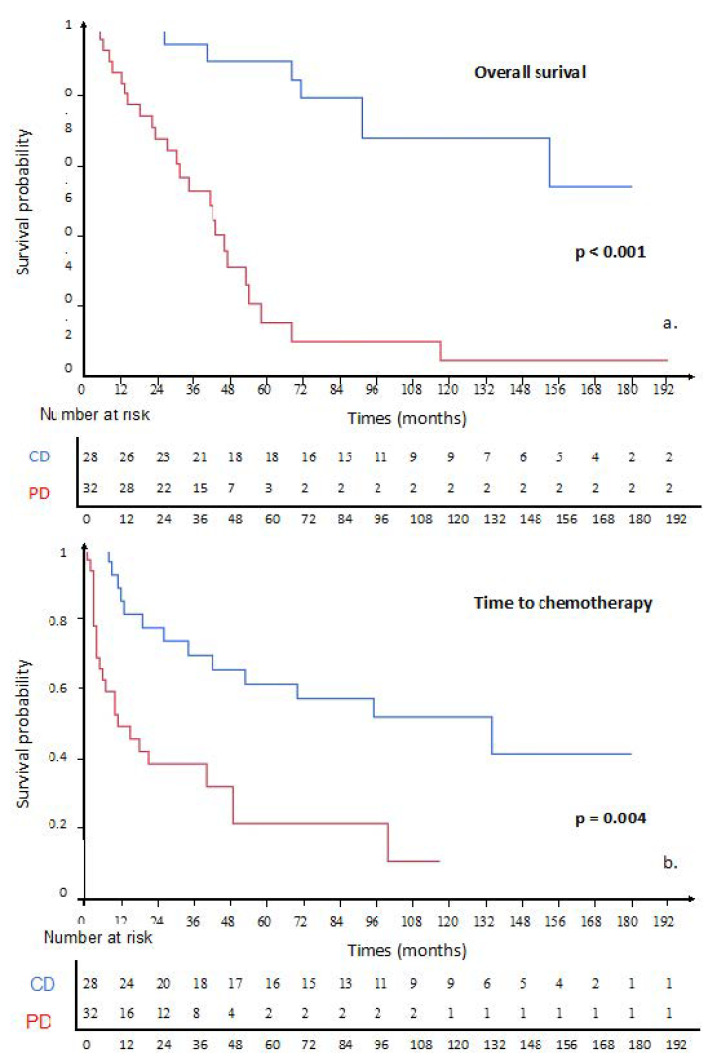
Kaplan-Meir estimate since mitotane initiation in DC patients and PD patient for (**a**). Overall survival; (**b**). Time to chemotherapy. (DC: controlled disease; PD: progressive disease.).

**Figure 3 cancers-14-02730-f003:**
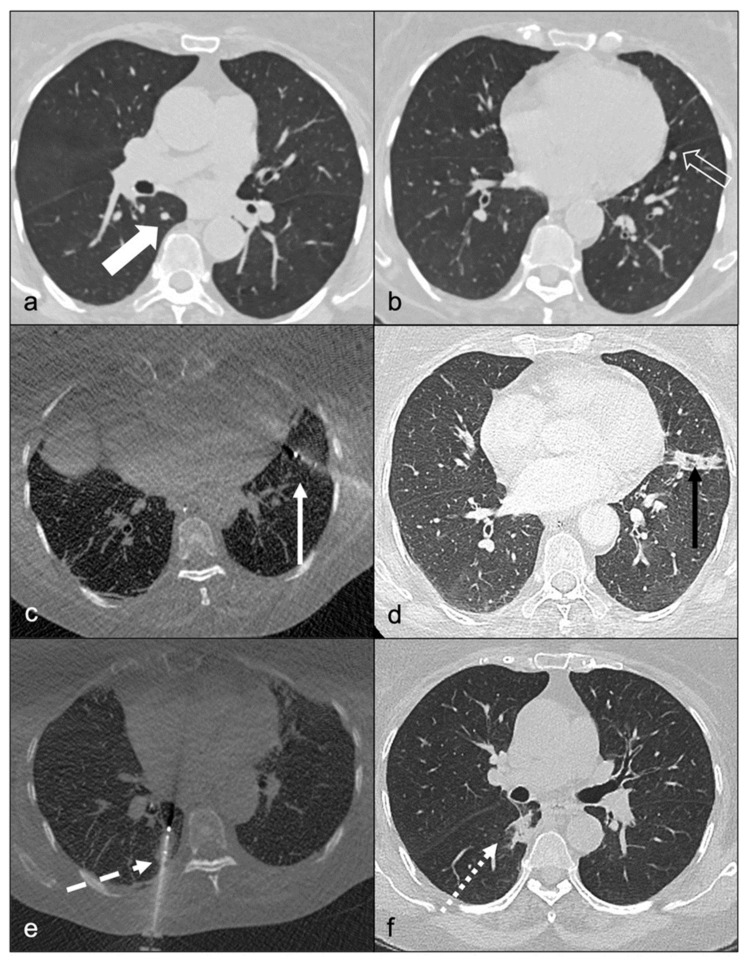
Complete response obtained with mitotane plus multiple locoregional treatments: a 65-year-old female patient with two small lung metastases in (**a**). right lower lobe and (**b**). oblique fissure. (**c**). Cryotherapy of the left fissure nodule complicated by a pneumothorax, which contraindicated right nodule treatment during the same intervention. (**d**). Left nodule sequel one month after cryotherapy with no residual pneumothorax; partial response obtained. (**e**). Right lower lobe nodule percutaneous treatment 2 months after the previous treatment. (**f**). Right nodule sequel one month after the second treatment; complete response was achieved.

**Figure 4 cancers-14-02730-f004:**
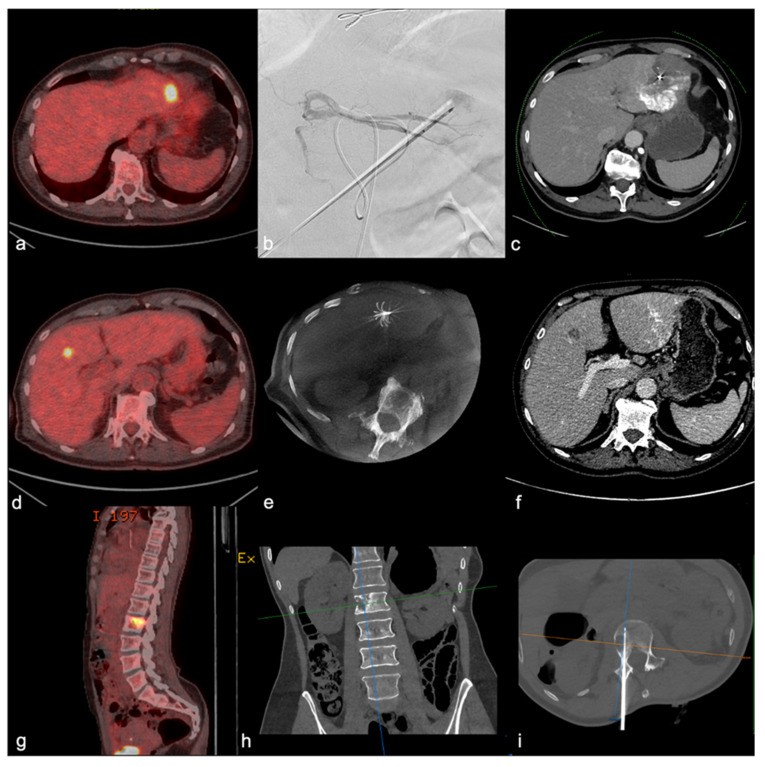
Iterative local treatments (LT): a 60-year-old woman with a history of stage IVa ACC with metachronous metastasis 14 months after mitotane initiation: (**a**). an axial PET-CT showing a two-centimeter FDG uptake within the third hepatic segment. (**b**). Fluoroscopy image of combined treatment with the association of chemoembolization and cryotherapy. (**c**). Six-week follow-up axial CT showing lipiodol uptake and no nodular contrast enhancement, in favor of complete response. (**d**). Axial PET-CT at 7 months of the first LT showing new centimetric FDG uptake within the fourth segment. (**e**). Radiofrequency ablation guided by ultrasound and CBCT with a deployable needle. (**f**). Contrast-enhanced follow-up CT at 6 weeks of the second LT showing unenhanced low-density ablation zone. (**g**). Axial PET-CT 6 months after the second LT (24 months from mitotane systemic treatment) depicting L2 nodular FDG uptake. (**h**). Non-contrast CT coronal view of a L2 lytic lesion. (**i**). Radiofrequency ablation of L2 vertebral body using a straight needle.

**Table 1 cancers-14-02730-t001:** Patient and tumor characteristics at stage IVa diagnosis.

Parameters	Patients
Patients charactristics	*n* = 60
Age at stade IV diagnosis (year, IQR)	48.1 (38.3–59.8)
<50	34 (57%)
≥50	26 (43%)
Gender	
Male	26 (43%)
Female	34 (57%)
Tumor characteristics	
Tumor related symptoms	
Y	43 (72%)
N	10 (16%)
Unknown	1 (2%)
R status in resected patients	
R0	22 (37%)
R1	7 (12%)
R2	0 (0%)
Rx	31 (52%)
Weiss score	
≤6	39 (68%)
>6	11 (32%)
Ki67%	
<20%	18 (34%)
≥20%	18 (25%)
Unknown	24 (41%)
Metastases (IVA)	
DFI	
Median time (months, IQR)	9.3 (0.8–22.0)
synchronous	21 (38%)
metachronous	37 (62%)
Lung	
Y	36 (60%)
N	24 (40%)
Liver	
Y	28 (47%)
N	32 (53%)
Nodes	
Y	8 (13%)
N	52 (87%)
Bone	
Y	4 (7%)
N	56 (93%)
Peritoneum	
Y	7 (12%)
N	53 (88%)
Local relapse	
Y	21 (35%)
N	39 (65%)
Oligometastasis	
Y	35 (58%)
N	25 (42%)
High tumor burden	
Y	37 (62%)
N	23 (38%)
*Metastatic organs*	
*n* = 1	41 (68%)
*n* > 1	19 (32)

Y: yes; N: no; IQR: interquartile range.

**Table 2 cancers-14-02730-t002:** Treatments and outcomes parameters.

Parameters	*n* (60) of Patients (%)
**Treatments**	
Adrenal space radiotherapy	14 (23%)
Second surgery	34 (57%)
Locoregional	25 (42%)
Hepatic	9 (15%)
Pulmonary	8 (13%)
Other	3 (5%)
Interventional radiology	35 (58%)
Cryotherapy	7 (12%)
Radiofrequency	18 (20%)
Microwaves	5 (8%)
Chemoembolization	20 (33%)
**Outcomes**	
Median follow-up (months)	104 (40–164)
Chemotherapy	
Within 6 month	8 (13%)
Overall	35 (58%)
Survival	
5-year OS	36 (60%)
Median OS (months)	68 (43–117)
Death	
Y	31 (52%)
N	22 (36%)
Lost to follow-up	7 (11%)

Y: yes; N: no; OS: overall survival.

**Table 3 cancers-14-02730-t003:** Characteristics of patients with disease control (best response during follow-up).

Parameters	Patients with PD	Patients with CD	*p*
Patients	*n* = 20	*n* = 40	
Age at stade IV diagnosis (year, IQR)	43.5 (31.7–59.2)	48.5 (39.3–61.2)	0.29
<50	11 (55%)	23 (57.5%)	
≥50	9 (45%)	17 (42.5%)	
Gender			
Male	7 (35%)	19 (47.5%)	0.41
Female	13 (65%)	21 (52.5%)	
*GRAS parameters*			>0.9
<1	2 (10%)	3 (7.5%)	
≥1	18 (90%)	37 (92.5%)	
DFI			
Median time (months, IQR)	9.0 (2.3–24.6)	11.5 (7–20.8)	0.68
Synchronous	7 (35%)	16 (40%)	
Metachronous	13 (65%)	24 (60%)	
Metastases (IVa)			
Lung			0.46
Y	13 (65%)	23 (57.5%)	
N	7 (35%)	17 (43.5%)	
Liver			0.41
Y	11 (55%)	17 (42.5%)	
N	9 (45%)	23 (57.5%)	
Nodes			0.70
Y	2 (10%)	6 (15%)	
N	18 (90%)	34 (85%)	
Bone			>0.9
Y	1 (5%)	3 (7.5%)	
N	19 (95%)	37 (92.5%)	
Peritoneum			0.40
Y	1 (5%)	6 (15%)	
N	19 (95%)	34 (85%)	
Local relapse			0.58
Y	8 (40%)	13 (32.5%)	
N	12 (60%)	27 (67.5%)	
Oligometastasis			
Y	6 (30%)	29 (72.5%)	0.002 *
N	14 (70%)	11 (27.5%)	
Dmax > 3 cm			
Y	3 (15%)	20 (50%)	0.011 *
N	17 (85%)	20 (50%)	
*Metastatic organ*			
*n* = 1	11 (55%)	30 (75%)	0.150
*n* > 1	9 (45%)	10 (25%)	

Y: yes; N: no; IQR: interquartile range; DC: controlled disease; PD: progressive disease; Dmax: maximum diameter; * *p*-value < 0.05.

**Table 4 cancers-14-02730-t004:** Univariate and multivariate analysis for Overall Survival (OS) and Time to second-line Treatment initiation (TTC).

Parameters	Univariate Analysis				Multivariate Analysis		
**OS**	Total (*n* = 60)	Hazard ratio	95% CI	*p*-value	Hazard ratio	95% CI	*p*-value
Sex							
Female	34	0.89	0.43–1.79	0.730	-	-	-
Male	26	1	-	-	-	-	-
GRAS factor							
*n* = 0	5	0.25	0.03–1.80	0.080	-	-	-
n > 0	55	1	-	-	-	-	-
Metastatic organ							
*n* = 1	41	0.30	0.13–0.64	0.003 *	0.31	0.14–0.69	0.005 *
*n* > 1	19	1	-	-	1	-	-
Oligometastatic							
Yes	25	0.37	0.18–0.77	0.008 *	0.40	0.19–0.82	0.014 *
No	35	1	-	-	1	-	-
Dmax < 3 cm							
Yes	37	0.49	0.22–1.04	0.055	-	-	-
No	23	1	-	-	-	-	-
DFI							
synchronous	23	0.86	0.39–1.73	0.610	-	-	-
metachronous	37	1	-	-	-	-	-
**TTC**							
Sex							
Female		1.20	0.60–2.37	0.600	-	-	-
Male		1	-	-	-	-	-
GRAS factor							
*n* = 0	5	0.90	0.27–2.97	0.870	-	-	-
n > 0	55	1	-	-	-	-	-
Metastatic organ							
n = 1	41	0.50	0.24–1.03	0.072	-	-	-
n > 1	19	1	-	-	-	-	-
Oligometastatic							
Yes	25	0.35	0.18–0.68	0.002 *	0.32	0.16–0.64	0.001 *
No	35	1	-	-	1	-	-
Dmax < 3 cm							
Yes		0.47	0.22–0.99	0.039 *	0.41	0.19–0.89	0.024 *
No		1	-	-	1	-	-
DFI							
synchronous	23	0.68	0.35 - 1.3	0.270	-	-	-
metachronous	37	1	-	-	-	-	-

OS: overall survival; TTC: time to chemotherapy; Y: yes; N: no; Dmax: maximum diameter; * *p*-value < 0.05; oligometastatic: <5 metastasis; Dmax: maximal diameter, DFI: disease-free interval: synchronous: <1 year; metachronous: ≥ 1 year.

## Data Availability

The data presented in this study are available on request from the corresponding author.

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
