# Peer review of "Loco-Regional Therapies in Oligometastatic Adrenocortical Carcinoma"

_cancers, 2022, doi:10.3390/cancers14112730_

Round 1
Reviewer 1 Report
I would like to thank the editor for letting me review this interesting manuscript, provided by a highy experienced team in the field of ACC.
Question about which metastatic patients (ie, with systemic disease) might benefit from loco-regional therapies is indeed crucial.
Moderate English changes and spell check is required in the manuscript. Moreover, abbreviations are not consistent in the manuscript: DC or CD (Figure 2) ; stages IVA-IVB-IVC or IVa-IVb-IVc, the latter being more understandable. If "low tumour burden ACC" is meant to be equivalent to "stage IVa ACC", consider one definition only.
Highlights: "DC is independantly correlated to OS…" needs modification since not all stage IVa ACC were included in the survival analysis.
Abstract: needs clarity and extensive Editing. LR should be precisely defined.
Introduction:
- "advanced ACC" should be clearly defined (lines 47 and 57)
- 31% of advanced ACC were stage IVa" (line 57) is a result and should not be presented in the Introduction paragraf
- The role of systemic chemotherapy should be stated, so that the readers (few are experts in the field of stage IV ACC) understand why assesment of TTC was chosen as a secondary endpoint.
- Last paragraph (lines 68-71) is unclear
Material and Methods:
- It is unclear why patients with stage IVa ACC managed with mitotane alone (19 patients) have not been included in the analysis ; it would be of interest to evaluate the DC, TTC and OS in these patients without LR (demonstrating the impact of M+LR over M alone) describing why LR could not be proposed.
- If all patients underwent surgery of the primary tumor, was LR using surgery a second surgery to treat recurrence or a single procedure associating resection of the primary tumor and abdominal metastases ? In other words, how many patients were stade IVa at diagnosis and how many presented metastatic recurrence during follow-up ? Please clarify.
- It is not clear wether patients with IVa ACC were initially placed under Mitotane before decision to perform LR ? In other words, was there a time interval for evaluation of progression under Mitotane before decision to perform LR ? Please state which specialists participate in the adrenal tumor board, clarify the choice for LR in stage IVa patients.
- If some patients included in the analysis were initially stage II-III ACC patients with metastatic recurrence, some might have been treated with mitotane as an adjuvant therapy. Please clarify.
- After defining metachronous /synchronous metastases, us these terms in the manuscript (and in Table 4).
- In the Statistical Analysis, Table 4 should not be presented as it is part of the Results; why Table 4 is presented before Table 1 ?
Results:
- Flow-chart is unclear (see above), as patients characteristics description in the text (lines 143-146); Flow-chat should describe "patients" and not "ACC"
- Presentation of all Tables needs revising
- Adrenal "lodge" is unappropriate (line 162 and table 2)
- Sentence in line 176 seems incomplete
Discussion:
- Lines 224-225 is part of the Results paragraph.
- Figure 3 in its present form is unnecessary; it could be used to present which patients are accessible to LR.
- Lines 282-287: please clarify.
Author Response
Reviewer #1
I would like to thank the editor for letting me review this interesting manuscript, provided by a highy experienced team in the field of ACC.
Question about which metastatic patients (ie, with systemic disease) might benefit from loco-regional therapies is indeed crucial.
Thank you for those extensive and precise reviews which improve our paper tremendously.
Q1.R1: Moderate English changes and spell check is required in the manuscript. Moreover, abbreviations are not consistent in the manuscript: DC or CD (Figure 2) ; stages IVA-IVB-IVC or IVa-IVb-IVc, the latter being more understandable. If "low tumour burden ACC" is meant to be equivalent to "stage IVa ACC", consider one definition only.
As asked, we modified our abbreviations to be consistent (CD, stage IVA, IVb, IVC are remplaced by DC, IVa, IVb, IVc) and we now use stage IVa ACC in place of low tumor burden.
Q2.R1: Highlights: "DC is independantly correlated to OS…" needs modification since not all stage IVa ACC were included in the survival analysis. OK, we changed for “DC is independently correlated to Overall Survival in stage IVa adrenocortical carcinoma treated by loco-regional treatments (LR)”.
Q3.R1: Abstract: needs clarity and extensive Editing. LR should be precisely defined. We edited the abstract according to your recommendations for it to be clearer and easily understandable.
Introduction:
Q4.R1: "advanced ACC" should be clearly defined (lines 47 and 57). Advanced ACC is defined by metastatic ACC. We added this information in the introduction.
Q5.R1: 31% of advanced ACC were stage IVa" (line 57) is a result and should not be presented in the Introduction paragraf. This is a result from Libé and al study. As it was nnot clear in our manuscript we slightly modified this sentence to be clearer.
Q6.R1: The role of systemic chemotherapy should be stated, so that the readers (few are experts in the field of stage IV ACC) understand why assesment of TTC was chosen as a secondary endpoint. Based on your suggestion we added: “Moreover, second line treatment is platin-based chemotherapy, that exhibit limited efficacy and no validated third line treatment” in Introduction.
Q7.R1 : Last paragraph (lines 68-71) is unclear. We modified this sentence for it to be more straightforward.
Material and Methods:
Q8.R1: It is unclear why patients with stage IVa ACC managed with mitotane alone (19 patients) have not been included in the analysis ; it would be of interest to evaluate the DC, TTC and OS in these patients without LR (demonstrating the impact of M+LR over M alone) describing why LR could not be proposed. We do agree with Reviewer #1. Our group recently published a study on this subject (Boileve and al(1)). It proves that in comparable population, patients that had LR and mitotane had better overall survival and TTC than patients with mitotane only. After this study our goal was to understand which patients will benefit the most of this strategy, hence this study.
Q9.R1: If all patients underwent surgery of the primary tumor, was LR using surgery a second surgery to treat recurrence or a single procedure associating resection of the primary tumor and abdominal metastases ? In other words, how many patients were stade IVa at diagnosis and how many presented metastatic recurrence during follow-up ? Please clarify. As stated in boileve and al paper, only 12% were stage IVa at ACC diagnosis. For patients not amendable to radical and total resection at initial diagnosis (stage IV patients), primary tumor resection is performed for tumor mass control and is associated with better prognosis. Metastasis are either treated by systemic treatments (stage IVb and IVc) or either by mitotane plus LR in stage IVa patients.
Q10.R1: It is not clear wether patients with IVa ACC were initially placed under Mitotane before decision to perform LR? In other words, was there a time interval for evaluation of progression under Mitotane before decision to perform LR ? Please state which specialists participate in the adrenal tumor board, clarify the choice for LR in stage IVa patients.
Specialists that participate to tumor board are neuroendocrine oncologists, radiologists radiotherapists, surgeons, interventional radiologists and nuclear physicians. At stage IVa diagnosis we decide either patients are accessible to LR plus mitotane or not and LR type was decided by consensus among physicians. Mitotane initiation was defined as the time of initiation of mitotane in synchronous metastatic patients, or as the time of stage IVa diagnosis in case of metachronous metastatic patients already under mitotane for adjuvant treatment. It is now stated in treatments and complications or in TTC and OS paragraphs.
Q11.R1: If some patients included in the analysis were initially stage II-III ACC patients with metastatic recurrence, some might have been treated with mitotane as an adjuvant therapy. Please clarify. We clarified this point in the previous answer (Q10.R1)
Q12.R1: After defining metachronous /synchronous metastases, us these terms in the manuscript (and in Table 4). Done
Q13.R1: In the Statistical Analysis, Table 4 should not be presented as it is part of the Results; why Table 4 is presented before Table 1? Thank you for this point. We displaced table 4 to its right position.
Results:
Q14.R1: Flow-chart is unclear (see above), as patients characteristics description in the text (lines 143-146); Flow-chat should describe "patients" and not "ACC". You are perfertly right and corrections has been made.
Q15.R1: Presentation of all Tables needs revising. Presentation has been improved as requested.
Q16.R1 : Adrenal "lodge" is unappropriate (line 162 and table 2). Replaced by adrenal space.
Q17.R1 : Sentence in line 176 seems incomplete. Correction has been done.
Discussion:
Q18.R1 : Lines 224-225 is part of the Results paragraph. It has been modified accordingly
Q19R1: Figure 3 in its present form is unnecessary; it could be used to present which patients are accessible to LR. This comment is relevant. Figure 3a is now used to illustrate why a delay is needed to reach best reponse to LR and Figure 3b is now used to show that iterative treatments can be performed along patient history.
Q19.R1: Lines 282-287: please clarify. This sentence was confusing and added no valuable information. We deleted it in this revised version of our study.

Reviewer 2 Report
Although this is a retrospective study, this is an important topic in a rare cancer and analyzes a significant cohort of patients with oligometastatic disease. The finding of tumor size(<3 cm) and number of tumors metastases (<5) may help better define the population of patients most likely to benefit from local therapy. This would be an important contribution to the literature.
There are some concerns including:
- There are several grammatical errors scattered throughout the manuscript such as misspelling of mitotane, missing periods or parentheses.
- Table formatting needs to be improved in presentation and size (Table 2 and 3).
- Tables are not in order as first Table in the manuscript sequence is Table 4.
- Some of the wording is confusing such as the statement at the end of the introduction, "The definition of oligo metastatic ACC......
- It would be helpful to clarify better with regard to the policy for obtaining informed consent for retrospective reviews. The current version indicates seems to imply that all patients were consented, although assuming there were a cohort of expired patients at the time of review.
- The paper might benefit from some discussion with regard to the current guidelines (such as NCCN ) which do support the treatment of oligometastatic with local therapies; and how this definition of oligo metastatic disease will help treating physicians.
Overall, this data would be a valuable contribution to the literature and main concerns are related to presentation.
Author Response
Reviewer #2
Although this is a retrospective study, this is an important topic in a rare cancer and analyzes a significant cohort of patients with oligometastatic disease. The finding of tumor size(<3 cm) and number of tumors metastases (<5) may help better define the population of patients most likely to benefit from local therapy. This would be an important contribution to the literature.
Thank you for this short and accurate sum-up of our study.
There are some concerns including:
Q1.R2: There are several grammatical errors scattered throughout the manuscript such as misspelling of mitotane, missing periods or parentheses. We corrected those errors and thank R#2 to have stressed this point.
Q2.R2: Table formatting needs to be improved in presentation and size (Table 2 and 3). Table formatting have been improved along R#2 remark.
Q3.R2: Tables are not in order as first Table in the manuscript sequence is Table 4. Table order has been corrected as asked.
Q4.R2: Some of the wording is confusing such as the statement at the end of the introduction, "The definition of oligo metastatic ACC...... This sentence has been deleted has it was not clear and not usefull in this part of text.
Q5.R2: It would be helpful to clarify better with regard to the policy for obtaining informed consent for retrospective reviews. The current version indicates seems to imply that all patients were consented, although assuming there were a cohort of expired patients at the time of review. In our group all ACC patients are proposed to sign informed consent to allow us to use their medical information for research purpose. They can retrieve this consent at any moment of the follow-up. Moreover, institutional ethics committee is asked for authorization for each retrospective study that we intend to do.
Q6.R2: The paper might benefit from some discussion with regard to the current guidelines (such as NCCN ) which do support the treatment of oligometastatic with local therapies; and how this definition of oligo metastatic disease will help treating physicians. We extended the paragraph about oligometastatic definition and preliminary studies. We added references about ESMO and NCCN recommendations in oligometastatic NSCLC.
Overall, this data would be a valuable contribution to the literature and main concerns are related to presentation.
Bibliography :
- Boileve A, Mathy E, Roux C, Faron M, Hadoux J, Tselikas L, et al. Combination of mitotane and locoregional treatments in low-volume metastatic adrenocortical carcinoma. J Clin Endocrinol Metab. 2021 Jun 18;dgab449.

Round 2
Reviewer 1 Report
The manuscript has been modified as requested.
Minor spell check are required :
line 22 : staged
line 24 : identified
lines 48 and 56 : not sure why you would prefer "Advanced" ACC over "metastatic" which seems more clear
line 78 : "stage" is present twice
Author Response
Reviewer #1
The manuscript has been modified as requested.
Minor spell checks are required:
Q1.R1: line 22: staged. Done
Q2.R1: line 24: identified. Modified as resquested.
Q3.R1: lines 48 and 56: not sure why you would prefer "Advanced" ACC over "metastatic" which seems more clear. We changed advanced ACC for metastatic ACC as suggested. We thank R#1 for this suggestion that clarified our message.
Q4.R1: line 78: "stage" is present twice. We slightly modified this sentence.
Reviewer #2
Q2.R2: Substantial correction and quality improvements has been made and feel this work would be a valuable contribution to the literature.
We would like to thanks both reviewers for their inputs that improved greatly our manuscript.

Reviewer 2 Report
Substantial correction and quality improvements has been made and feel this work would be a valuable contribution to the literature.
Author Response
Reviewer #1
The manuscript has been modified as requested.
Minor spell checks are required:
Q1.R1: line 22: staged. Done
Q2.R1: line 24: identified. Modified as resquested.
Q3.R1: lines 48 and 56: not sure why you would prefer "Advanced" ACC over "metastatic" which seems more clear. We changed advanced ACC for metastatic ACC as suggested. We thank R#1 for this suggestion that clarified our message.
Q4.R1: line 78: "stage" is present twice. We slightly modified this sentence.
Reviewer #2
Q2.R2: Substantial correction and quality improvements has been made and feel this work would be a valuable contribution to the literature.
We would like to thanks both reviewers for their inputs that improved greatly our manuscript.
Best regards,
